

# Transcriptome analysis unveils the mechanisms of lipid metabolism response to grayanotoxin I stress in *Spodoptera litura*

Yi Zhou[1,*], Yong-mei Wu[1,*], Rong Fan[1], Jiang Ouyang[1], Xiao-long Zhou[1], Zi-bo Li[1], Muhammad Usman Janjua[2], Hai-gang Li[1,3], Mei-hua Bao[1,3] and Bin-sheng He[1]

[1] Changsha Medical University, The Hunan Provincial Key Laboratory of the TCM Agricultural Biogenomics, Changsha, Hunan, China
[2] Changsha Medical University, School of International Education, Changsha, Hunan, China
[3] Changsha Medical University, Hunan Key Laboratory of the Research and Development of Novel Pharmaceutical Preparations, School of Pharmaceutical Science, Changsha, Hunan, China
[*] These authors contributed equally to this work.

Corresponding authors
Mei-hua Bao, mhbao78@163.com
Bin-sheng He, hnaios@163.com

## ABSTRACT

**Background.** *Spodoptera litura* (tobacco caterpillar, *S. litura*) is a pest of great economic importance due to being a polyphagous and world-distributed agricultural pest. However, agricultural practices involving chemical pesticides have caused resistance, resurgence, and residue problems, highlighting the need for new, environmentally friendly methods to control the spread of *S. litura*.

**Aim.** This study aimed to investigate the gut poisoning of grayanotoxin I, an active compound found in *Pieris japonica*, on *S. litura*, and to explore the underlying mechanisms of these effects.

**Methods.** *S. litura* was cultivated in a laboratory setting, and their survival rate, growth and development, and pupation time were recorded after grayanotoxin I treatment. RNA-Seq was utilized to screen for differentially expressed genes (DEGs). Gene Ontology (GO) and Kyoto Encyclopedia of Genes and Genomes (KEGG) pathway enrichment analyses were conducted to determine the functions of these DEGs. ELISA was employed to analyze the levels of lipase, 3-hydroxyacyl-CoA dehydrogenase (HOAD), and acetyl-CoA carboxylase (ACC). Hematoxylin and Eosin (H & E) staining was used to detect the development of the fat body.

**Results.** Grayanotoxin I treatment significantly suppressed the survival rate, growth and development, and pupation of *S. litura*. RNA-Seq analysis revealed 285 DEGs after grayanotoxin I exposure, with over 16 genes related to lipid metabolism. These 285 DEGs were enriched in the categories of cuticle development, larvae longevity, fat digestion and absorption. Grayanotoxin I treatment also inhibited the levels of FFA, lipase, and HOAD in the hemolymph of *S. litura*.

**Conclusion.** The results of this study demonstrated that grayanotoxin I inhibited the growth and development of *S. litura*. The mechanisms might, at least partly, be related to the interference of lipid synthesis, lipolysis, and fat body development. These findings provide valuable insights into a new, environmentally-friendly plant-derived insecticide, grayanotoxin I, to control the spread of *S. litura*.

## INTRODUCTION

*Spodoptera litura* (*S. litura*), also named tobacco cutworm pest, is a polyphagous and widely distributed agricultural pest that causes damage to over 300 host plants. It is found in Africa, the Middle East, Southern Europe, and Asia (*Prajapati, Varma & Vadassery, 2020*). Currently, the control of *S. litura* relies heavily on chemical pesticides. However, a new environmentally friendly methods is urgently needed due to the resistance, resurgence, and residue problems caused by unreasonable long-term use of chemical pesticides (*Xu et al., 2020b*).

One promising approach for developing environmentally friendly pesticides is screening bioactive compounds from natural plant products. Compared to synthetic chemical insecticides, botanical insecticides have been considered to have low environmental and mammalian risk, high specificity and safety, low risk of resistance development, and low environmental persistence (*Seiber et al., 2014*; *Regnault-Roger, Vincent & Arnason, 2012*; *Isman & Grieneisen, 2014*). Several classes of molecules derived from plant products were demonstrated to be bioactive, such as terpenes, flavonoids, alkaloids, and polyphenols (*Deota & Upadhyay, 2005*; *Souto et al., 2021*). These plant-derived insecticides achieved their effects through mechanisms of affect the nervous system, respiratory and endocrine systems, as well as water balance in insects (*Souto et al., 2021*). For example, azadirachtin is a series of tetracyclic triterpenoid compounds extracted from plant *Azadirachta indica A. Juss.* It achieved insecticidal effects by deterring feeding, interfering with egg laying, disrupting insect metamorphosis, repelling larvae, and inhibiting their growth (*Sun et al., 2022b*; *Yu et al., 2023*). Rotenone induced insect cell necrosis *via* cytoplasmic membrane damage and mitochondrial dysfunction (*Sun et al., 2021*). Pyrethrins kill mosquitos through modulating voltage-gated sodium channels (*Du et al., 2013*). Triterpenoids extracted from plants are an important class of compounds extensively studied in the research of plant-based pesticides (*Pavela et al., 2019*). Grayanotoxin I is a diterpenoid belonging to the grayanotoxin family. Grayanotoxins are commonly found in plants of the *Ericaceae* family, including *Rhododendron* and *Pieris japonica* (*Yao et al., 2006*). *Pieris japonica* has been reported to have anti-insect effects (*Xie, 2009*). As one of the most abundant and potent toxins in *Pieris japonica*, grayanotoxin was shown to interact with voltage-gated sodium channels, lead to the disruption of neuronal signaling, and cause symptoms such as dizziness, analgesic, weakness, and cardiac effects when ingested (*Zheng et al., 2020*). However, the precise effects and mechanisms of grayanotoxin I on agricultural pests are still largely unknown. Our preliminary studies showed that grayanotoxin I significantly inhibited the growth and development of *S. litura*. To further explore the mechanisms of this effect, the present study screened the transcriptome of *S. litura*, analyzed the functions of differentially expressed genes (DEGs), detected changes in the development of the fat body, and measured the levels of free fatty acids (FFA), 3-hydroxyacyl-CoA dehydrogenase (HOAD), Acetyl-CoA carboxylase (ACC), and lipase after grayanotoxin I treatment. The

present study aims to shed light on the effects and mechanisms of grayanotoxin I on *S. litura* and contribute to the development of new environmentally friendly pesticides.

## MATERIALS & METHODS

### Materials and reagents

Grayanotoxin I was procured from Sichuan Biocrick Biotech Co. Ltd (4720-09-6, Chengdu, China). The free fatty acid assay kit was obtained from Jiancheng Co. Ltd. (Nanjing, China), while the hematoxylin-eosin (H & E) staining solution was obtained from Beyotime Biotechnology (Shanghai, China). The Vazyme® HiScript III 1st Strand cDNA Synthesis Kit (+gDNA wiper) and Vazyme ChamQ Universal SYBR qPCR Master Mix were purchased from Vazyme Corporation (Nanjing, China). The primers were synthesized by Takara (Dalian, China). Further, Lipase (JM-00078O1), 3-hydroxyacyl-CoA dehydrogenase (JM-00048O1), and Acetyl-CoA carboxylase (JM-00064O1) ELISA kits were procured from Jingmei Biotechnology (Jiangsu, China).

### *Spodoptera litura* culture, treatment, and sample collection

The larvae of *S. litura* were obtained from Keyun Biopesticide Co. Ltd in Henan, China. These larvae were sourced from fields free from heavy metal pollution with no prior application of chemical insecticides. Optimal laboratory culture conditions of a temperature of $25 \pm 2\,°C$, humidity of 75%–85%, and a light cycle of light/dark: 14 h/10 h were employed for the rearing of the larvae. Only the second instar larvae with uniform size and normal development were selected for further testing.

To investigate the effects of grayanotoxin I on *S. litura*, the plant-derived insecticide, matrine was used as the positive control. Matrine is an alkaloid derived from plants belonging to the Sophora genus. As a naturally occurring plant-based pesticide, matrine generally poses low toxicity to humans. Matrine operates as a broad-spectrum insecticide, effectively targeting pests through both contact and ingestion mechanisms. The second instar larvae were randomly divided into the normal diet, different concentrations of grayanotoxin I-containing diet, or matrine-containing diet group. The diets were prepared by adding 7 mL of ddH$_2$O, 1.25–6.25 mg/L grayanotoxin I, or 0.4% matrine solution to 5 g diet. The survival rates were calculated at 24-hour, 48-hour, and 72-hour treatments. The midgut of *S. litura* larvae fed on a 1.25% grayanotoxin I-contained diet or normal diet (ddH$_2$O) for 72 h was collected for RNA-Seq.

For analysis of body weight and developmental time, sublethal concentrations (0.62–1.25 mg/L) of grayanotoxin I were used to treat *S. litura* larvae. The diets were prepared by adding 7 mL of grayanotoxin I solution to 5 g of normal diet. The wet body weight of each larvae was collected at each instar stage until pupation, and the data was recorded.

### Hematoxylin and Eosin (H & E) staining of fat body

The growth rate of insects is largely regulated by the fat body (*Yuan et al., 2020*). To assess the development of this crucial tissue, we utilized the H & E staining method, as previously described (*Yamahama, Seno & Hariyama, 2008*). The specimens were subjected to a 5-hour incubation at 5 °C in 10% sucrose in 0.01 M phosphate-buffered saline (PBS, pH 7.4),

with sucrose concentration gradually increased to 20%. The samples were then embedded in an optimal cutting temperature (OCT) compound and instantaneously frozen with dry ice. Further, frozen samples were sectioned at 10 μm and stained by the H & E method to obtain images. The images were examined under a microscope to evaluate the development of the fat body.

## RNA extraction and RNA-sequencing

To further explore the impact of grayanotoxin I on the expression of lipid metabolism-related genes, RNA-Sequencing using Illumina NovaSeq 6000 platform (Illumina, San Diego, CA, USA) was carried out at Shanghai Personal Biotechnology Cp., Ltd (Shanghai, China). The methodology was consistent with previously published studies (*Bao et al., 2016a*; *Bao et al., 2016b*). Briefly, total RNA was extracted using the Trizol reagent. The quality and quantity of total RNA were assessed by measuring the absorbance on wavelengths of 260 nm and 280 nm by NanoDrop spectrophotometer (Thermo Scientific, Waltham, MA, USA). After the removal of rRNA by using poly-T oligo-attached magnetic beads, the total RNA was fragmented by using divalent cations under elevated temperature in an Illumina proprietary fragmentation buffer. The first strand cDNA was synthesized using random oligonucleotides and Super Script II. Subsequently, the second strand cDNA synthesis was performed by using DNA Polymerase I and RNase H. For hybridization preparation, the DNA fragments' 3′ ends were adenylated, followed by ligation of Illumina PE adapter oligonucleotides. To obtain cDNA fragments of the desired length (400–500 bp), the library fragments were purified using the AMPure XP system (Beckman Coulter, Pasadena, CA, USA). DNA fragments possessing adapter molecules on both ends were selectively enriched through a 15-cycle PCR reaction with the Illumina PCR Primer Cocktail. The resulting products were purified using the AMPure XP system and the quantity was measured using the Agilent high-sensitivity DNA assay on a Bioanalyzer 2100 system (Agilent, Santa Clara, CA, USA). Finally, the sequencing library was sequenced on the NovaSeq 6000 platform (Illumina, San Diego, CA, USA) by Shanghai Personal Biotechnology Cp. Ltd.

## Differentially expressed genes (DEGs) identification

The reference genome used in the present transcriptome was ASM270686v3. The sequencing data was filtered to get high-quality sequences by using Cutadapt (v1.15) software. The filtered data were mapped to the reference genome using HISAT2 (v2.0.5). The analysis of *S. litura* mRNA expression was performed using HTSeq (0.9.1) statistics. The original expressed read count value per gene was normalized *via* the FPKM method. DESeq (1.30.0) was employed to analyze differences in mRNA expression levels. RNAs with $|\log2\text{FoldChange}| > 1.0$ and $P$-value $< 0.05$ were identified as differentially expressed. To perform heatmap clustering, MeV 4.9.0 software was used. Using this method, differentially expressed lipid metabolism-related genes were selected and heatmap clustering was conducted.

**Table 1  The primers used in the present analysis.**

| Gene name | Forward primer | Reverse primer | Product lenghth | Amplified gene regions |
|---|---|---|---|---|
| Phospholipase A1-like | TCCTTGTCCACTCAGATATGT | GTTGATAACCGTGCGATGTA | 102 bp | Coding region |
| Acyl-CoA reductase | CTGGTTGATGCTCTGCTGTT | TGCCATTCCTTCGTTGTGTAAT | 113 bp | Coding region |
| Acyl-CoA desaturase | GCTTCTTCTTCTGCCACATC | ACATCACCATCCAATCACCTT | 111 bp | Coding region |
| Fatty acid-binding protein 2 like | TTCCTTAACAAGAACTACAA | AGTATCTCCATCCTTAGTC | 138 bp | Coding region |
| $\beta$-actin | GCATCCACGAGACCACTTACAA | CTGTGTTGGCGTACAAGTCCTTA | 75 bp | Coding region |
| GAPDH | GGGTATTCTTGACTACAC | CTGGATGTACTTGATGAG | 184 bp | Coding region |

### RT-qPCR verification of lipid metabolism-related DEGs

To verify the expression of four differentially expressed lipid metabolism-related genes, we utilized RT-qPCR as described previously (*Bao et al., 2018*). Total RNAs were extracted using Trizol reagent, followed by reverse transcription to cDNA utilizing the Vazyme HiScript III 1st Strand cDNA Synthesis Kit (+gDNA wiper). PCR reactions were carried out using the Vazyme ChamQ Universal SYBR qPCR Master Mix kit on the Applied Biosystems Quantstudio 5 system. The qPCR program was 95 °C for 30 s, followed by 40 cycles of 95 °C for 5 s, and 60 °C for 30 s. The GAPDH and $\beta$-actin were used as reference genes. The primers were presented in Table 1. The non-transcribed RNA was used as a negative control. The melting curve analysis was performed to verify the specificity of PCR products. All samples were run in triplicate and analyzed using the $2^{(-\Delta\Delta Ct)}$ method.

### Detection of FFA, lipase, ACC, and HOAD

Lipase, HOAD, and ACC are enzymes that play key roles in the metabolism of fatty acids. To investigate the impact of grayanotoxin I on the lipid metabolism of *S. litura*, we employed an ELISA-based approach to measure the levels of lipase, HOAD, and ACC in the hemolymph of 5th instar larvae. Briefly, hemolymph samples were collected in a 1.5 ml tube with 0.1% dithiothreitol (DTT), and centrifuged for 5 min (10,000 rpm) at 4 °C. The supernatant was stored at −80 °C for further use (*Bai & Grewal, 2007*). The ELISA analysis was conducted according to the manufacturer's instructions. Specifically, 50 μL of serum samples were added to enzyme-linked immunosorbent plates, mixed with enzyme labeling reagents, and incubated at 37 °C for 60 min. The liquid was then removed, and each well was washed five times with washing solution before adding chromogenic reagent and mixing. The mixture was incubated for 15 min at 37 °C in the dark, after which the reaction was halted using a stop solution. The absorbance value was then measured to determine the levels of lipase, HOAD, and ACC.

FFA was measured by using the fatty acid assay kit purchased from Jiancheng Co. Ltd. (Nanjing, China) according to the manufacturer's instructions. The assay kit is based on the principle that FFA reacts with copper ions to form fatty acid copper salts, which are soluble in chloroform. By using the copper reagent method to determine the copper ion content, the content of FFA can be estimated by colorimetric assay.

### Gene ontology (GO) enrichment and kyoto encyclopedia of genes and genomes (KEGG) and protein-protein interaction (PPI) analysis of lipid metabolism-related differentially expressed genes

To investigate the functions of differentially expressed genes related to lipid metabolism, we conducted GO enrichment analysis and KEGG pathway analysis. This analysis was carried out using the online tool DAVID (https://david.ncifcrf.gov/) (*Yu et al., 2022*; *Xu et al., 2020a*). The top 10 terms from biological process (BP), cell components (CC), molecular function (MF), and KEGG pathway were visualized, and a *P*-value < 0.05 was considered significant for both GO terms and KEGG pathways.

To further examine the interactions between lipid metabolism-related DEGs, we utilized the online tool STRING (https://string-db.org/). As *S. litura* data was not available in STRING, we used *Bombyx mori* data as an alternative. We also performed a further analysis of the signal pathways of the lipid metabolism-related DEGs on the KEGG pathway (https://www.genome.jp/kegg/).

### Statistical analysis

All the statistics were presented in the form of mean $\pm$ S.D. The significance of the differences was analyzed by ANOVA followed by the Newman-Student-Keuls test. A value of $P < 0.05$ was considered statistically significant.

## RESULTS

### Influence of grayanotoxin I on *S. litura* growth and development

To investigate the impact of grayanotoxin I on *S. litura*, we monitored the survival rate, growth, and development of the insects after being subjected to grayanotoxin I-contained, matrine-contained, or normal diet. As depicted in Fig. 1A, the application of a positive control, 0.4% matrine, reduced the survival rate to 18.8% after a 72-hour treatment. While 72-hour treatment with 6.25 mg/L grayanotoxin I reduced the survival rate to 40.0%, as compared to the normal diet (ddH$_2$O, survival rate of 96.7%). Additionally, lower concentrations of grayanotoxin I (0.62–1.25 mg/L) significantly hindered the growth of *S. litura* (Figs. 1B–1C). Compared to the ddH$_2$O group, the 0.2% matrine hindered the 95.3% body weight of *S. litura* on day 14. The suppression rate was 90.65% for 1.25 mg/L grayanotoxin I, and 56.29% for 0.65 mg/L grayanotoxin I after 14-day treatment (Figs. 1B–1C). Furthermore, we observed a significant delay in the pupation time of *S. litura* because of grayanotoxin I (Fig. 1D). The average pupation time for the ddH$_2$O group was 14.72 days. While it was 20.23 days for 1.25 mg/L grayanotoxin I treatment and 18.25 days for 0.62 mg/L grayanotoxin I treatment (Fig. 1D).

### Inhibition effect of grayanotoxin I on *S. litura* fat body development

In the present study, H & E staining was conducted to investigate the relationship between fat body development and the growth of *S. litura*. As illustrated in Fig. 2, a noticeable accumulation of fat in the fat body was observed in the ddH$_2$O control group (Fig. 2A). However, treatment with grayanotoxin I resulted in a significant inhibition of fat body development (Fig. 2B).

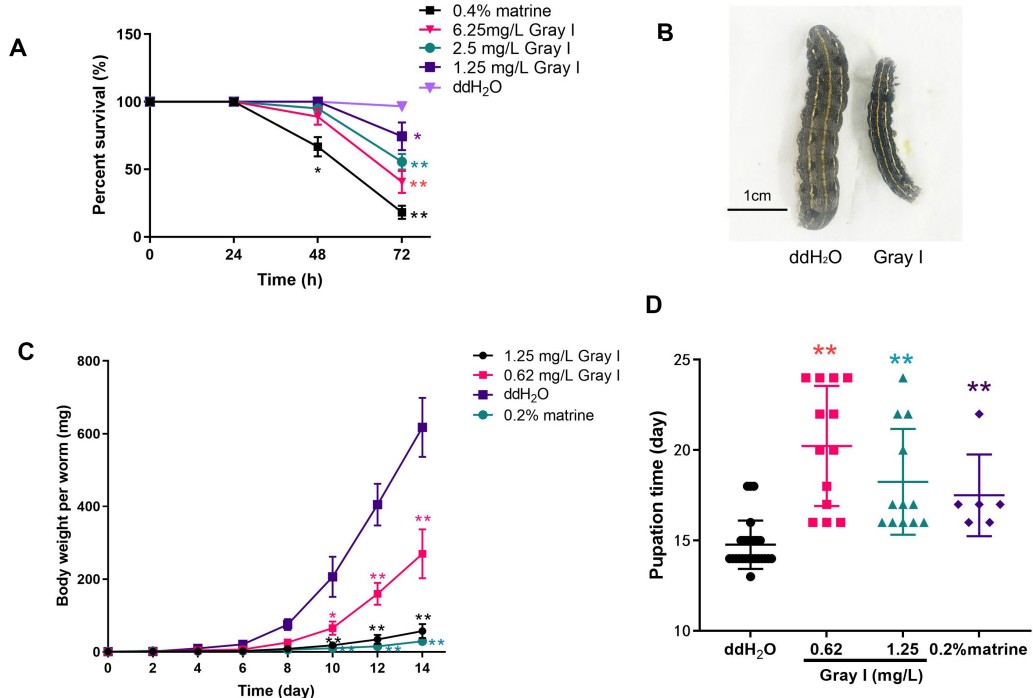

**Figure 1** **Effects of grayanotoxin I on survival rate, the growth & development of *S. litura*.** The second instar larvae of *S. litura* were fed with the normal diet, grayanotoxin I-containing diet, or matrine-containing diet. The survival rate, body length, body weight, and pupation time were measured. (A) The survival rate of *S. litura* after ddH$_2$O, 1.25–6.25 ml/L grayanotoxin I, or 0.4 % matrine treatment in 24, 48, and 72 hours; (B) the body length of *S. litura* between ddH$_2$O or 0.62 mg/L grayanotoxin I treatment on day 14; (C) the body weight-time curve after 0.62–1.25 ml/L grayanotoxin I, ddH$_2$O, or sublethal matrine (0.2 %) treatment; the body weight of each larvae was measured every 2 days. (D) The pupation time after grayanotoxin I, ddH$_2$O, or sublethal matrine (0.2 %) treatment. All data were presented in mean ± SD, **$P < 0.01$; *$P < 0.05$ *vs* ddH$_2$O group.

## Gene expression profiles of *S. litura* under grayanotoxin I treatment

To investigate the mechanisms of grayanotoxin I, we analyzed the transcriptome alteration after 72-hour 1.25% grayanotoxin I treatment by using the RNA-Seq method. The statistical power of this RNA-Seq data calculated in "RNASeqPower" was 0.855 (sequencing depth: 60, sample size: 3). As a result, 285 DEGs were identified. Among them, 151 were upregulated and 134 were downregulated (Figs. 3A–3B).

## GO and KEGG enrichment of differentially expressed lipid metabolism-related genes

To get further insight into the functions of the 285 DEGs, we carried out KEGG pathway enrichment and GO enrichment analysis. In the GO enrichment analysis, these DEGs were mostly enriched in the MF terms related to the structural constituents of chitin-based cuticle; BP terms associated with cuticle development; and CC terms related to the extracellular matrix (as depicted in Fig. 3C). The KEGG analysis (Fig. 3D) revealed that these DEGs were enriched in several pathways including the organismal system terms of longevity regulating pathway, cytosolic DNA-sensing pathway, and fat digestion and

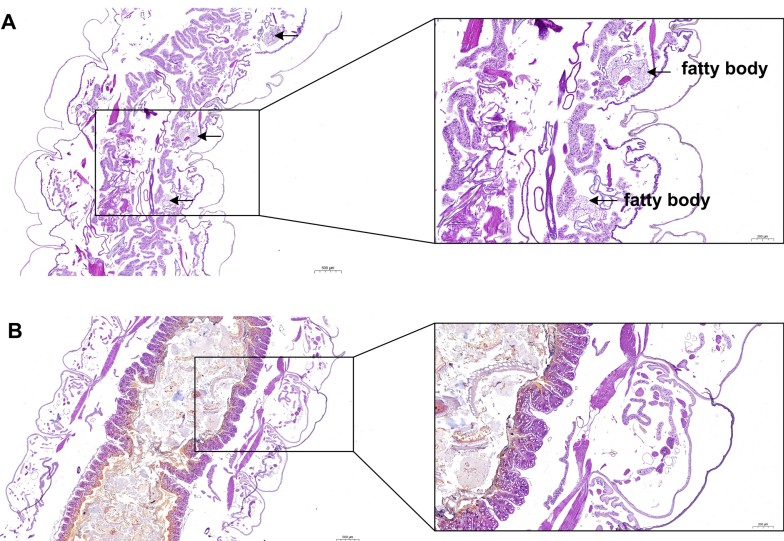

**Figure 2** **The development of fatty body after treatment of grayanotoxin I.** After treatment with grayanotoxin I for 14 days, the larvae of *S. litura* specimens were sectioned and stained by Hematoxylin and Eosin. The images were examined under a microscope to evaluate the development of the fat body. A, *S. litura* treated by ddH$_2$O; B, *S. litura* treated by grayanotoxin I.

absorption pathway; the metabolism terms of cutin, suberin, wax biosynthesis, linoleic acid metabolism, insect hormone biosynthesis, and unsaturated fatty acid synthesis; the cellular process terms of peroxisome.

## The effects of grayanotoxin I on lipid metabolism-related gene profile expression, lipid metabolism-related enzyme activities in the hemolymph, and FFA level in *S. litura*

In our RNA-Seq analysis, we discovered many DEGs related to lipid metabolism. Specifically, we observed an upregulation of genes such as acyl-CoA desaturase, esterase E4, and phospholipase, and downregulated genes such as fatty acid elongase, fatty acid-binding protein, and pancreatic-like lipase following treatment with grayanotoxin I (Fig. 4A). The results of RNA-Seq were verified by qPCR analysis, which was shown in Fig. 4B.

Besides, grayanotoxin I (1.25 mg/L) treatment dramatically decreased the level of FFA in the hemolymph of *S. litura* (Fig. 4C). Further ELISA analysis revealed a significant decrease in lipase and HOAD mRNA levels after treatment with grayanotoxin I, compared to the normal group ($P < 0.05$). A slight decrease in ACC mRNA was also found after grayanotoxin I treatment (Figs. 4D–4F).

## PPI analysis of lipid metabolism-related DEGs analysis

The PPI of the lipid metabolism-related genes was shown in Fig. 5A. Red circles were upregulated genes in *S. litura* after grayanotoxin I treatment, while green circles were downregulated genes.

The LOC111354773 (putative fatty acyl-CoA reductase), LOC111355891 (acyl-CoA desaturase 1-like), LOC111350394 (ELOVL fatty acid elongase), LOC111349277

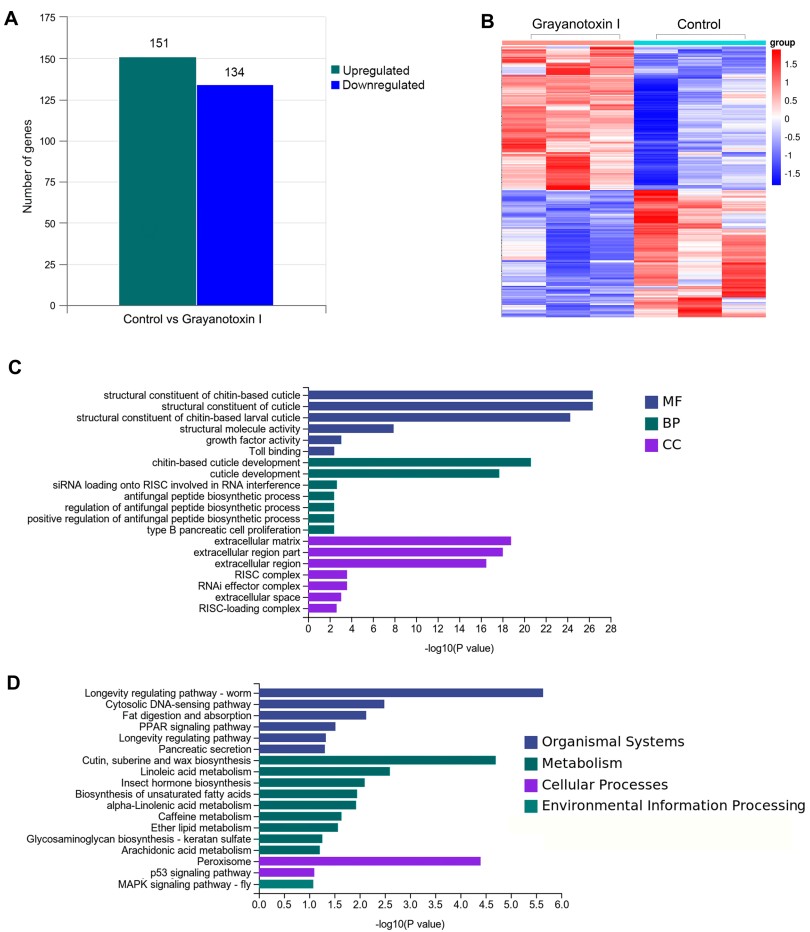

**Figure 3** **The transcriptomic analysis, GO enrichment, and KEGG enrichment of differentially expressed genes after grayanotoxin I treatment in *S. litura*.** (A) The number of upregulated and downregulated genes after grayanotoxin I treatment; (B) the heatmap of all differentially expressed genes after grayanotoxin I treatment; (C) GO enrichment of differentially expressed genes; (D) KEGG enrichment of differentially expressed genes.

(elongation of very long chain fatty acids protein 7 like), and LOC111360381 (fatty acid-binding protein 2 like) were connected clearly in the network.

Further analysis revealed that LOC111354773 (putative fatty acyl-CoA reductase), LOC111355891 (acyl-CoA desaturase 1-like), LOC111355893 (acyl-CoA desaturase 1-like), LOC111352061 (putative fatty acyl-CoA reductase), and LOC111356581 (fatty acyl-CoA reductase wat-like) were enriched in the longevity regulating pathway and were relevant to the aging of the larvae. The aforementioned genes along with LOC111350394 (ELOVL fatty acid elongase), LOC111348151 (phospholipase A1-like), and LOC111356581 (fatty acyl-CoA reductase wat-like) were found to be associated with lipid metabolism. Additionally, LOC111355891 (acyl-CoA desaturase 1 like), LOC111360381 (fatty acid-binding protein 2 like), and LOC111355893 (acyl-CoA desaturase 1-like) were found to be relevant to the PPAR signaling pathway, as documented in Table 2 and Fig. 5B.

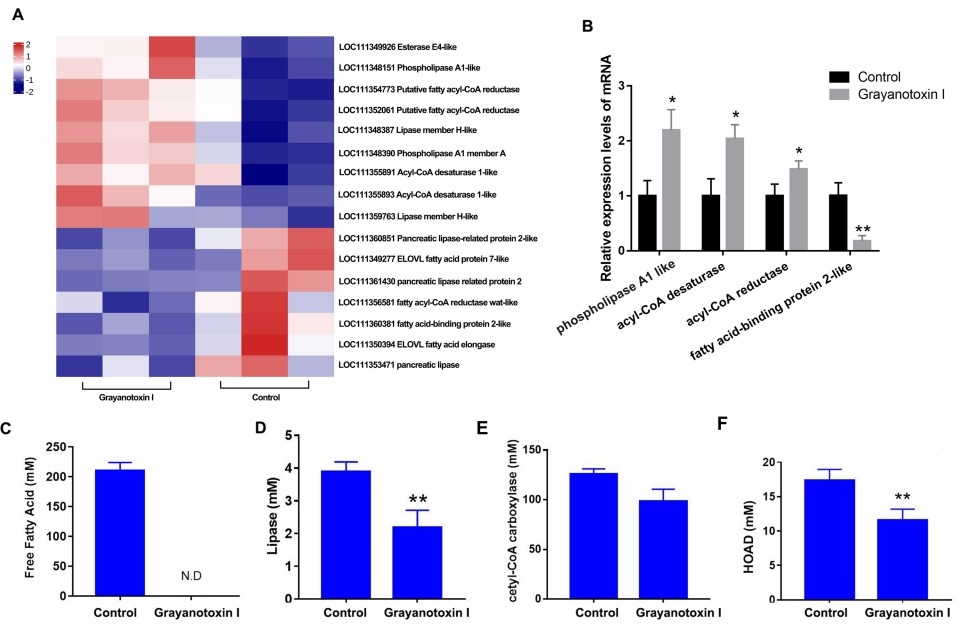

**Figure 4** **Effects of grayanotoxin I on lipid metabolism-related genes, lipid metabolism-related enzyme activities, and FFA levels in *S. litura*.** The second instar larvae of *S. litura* were treated with ddH$_2$O (control group) or 1.25 mg/L grayanotoxin I-containing diet for 72 h following which the midgut of *S. litura* was collected for RNA-Seq. (A) The heatmap of differentially expressed lipid metabolism-related genes; (B) qPCR verification of 4 randomly chosen lipid metabolism-related genes; (C–F) the level of free fatty acid, lipase, acetyl-CoA carboxylase, and HOAD in the hemolymph of *S. litura*. All data were presented in mean ± SD, *$P < 0.05$, **$P < 0.01$ *vs.* control group.

# DISCUSSION

The impact of grayanotoxin I on *S. litura* was evaluated in the present study, revealing a significant reduction in the survival rate, larvae growth, and delayed pupation. Transcriptome analysis identified 285 DEGs responding to grayanotoxin I treatment. GO enrichment and KEGG pathway enrichment indicated grayanotoxin I affected the expression of genes related to cuticle development, extracellular matrix, wax biosynthesis, insect hormone biosynthesis, fat digestion and absorption, *etc*. Notably, over sixteen of these DEGs were linked to lipid metabolism, with a significant decrease in FFA, lipase, and HOAD levels. These findings implicated grayanotoxin I probably interfered in lipid synthesis, lipolysis, lipid trafficking, and fat body development, ultimately restraining the growth of *S. litura*.

Traditional Chinese Medicine (TCM) has long been recognized for its low resistance and high efficiency, making it a popular remedy for a wide range of human diseases as well as agricultural insect infestations (*Deota & Upadhyay, 2005*; *Wang et al., 2022*; *Wei et al., 2018*; *Wang et al., 2016*). Grayanotoxin I is a diterpenoid belonging to the grayanotoxin family. Grayanotoxins are commonly found in plants of the *Ericaceae* family, including *Rhododendron* and *Pieris japonica* (*Yao et al., 2006*). Previously, grayanane diterpenoid glucosides were recognized as potent analgesics (*Zheng et al., 2020*). Our study found

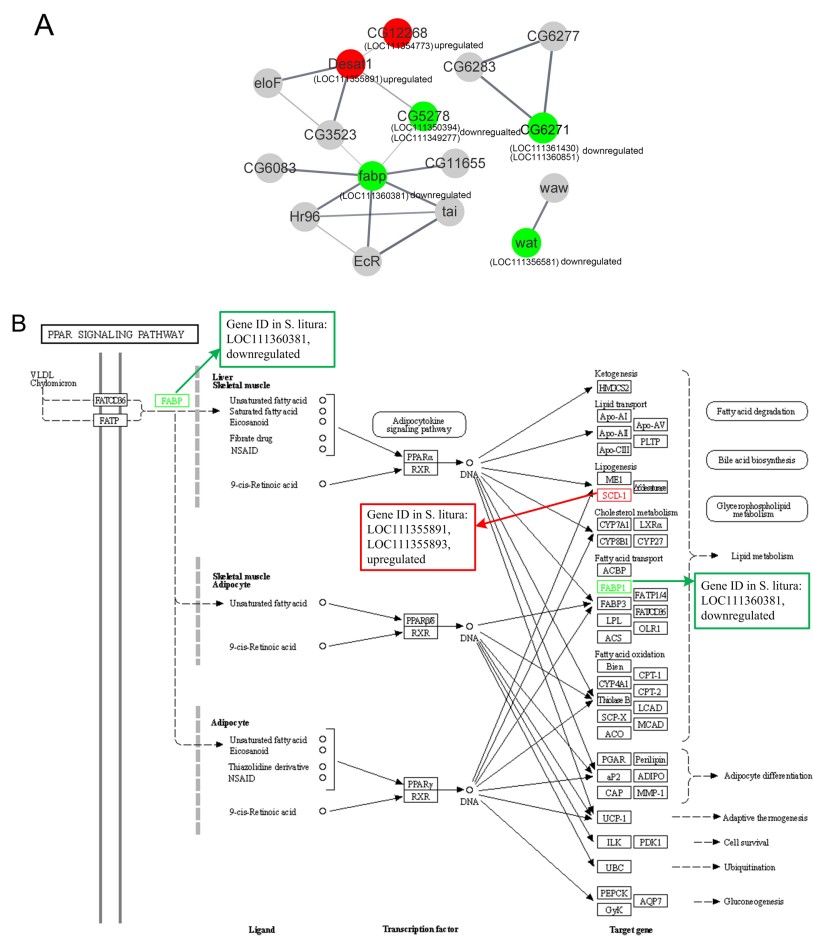

**Figure 5** **The protein-protein interactions and signal pathways of lipid-related DEGs.** (A) The protein–protein interaction of lipid-related DEGs analyzed by STRING online software; (B) the visualization of the PPAR signaling pathway obtained from KEGG pathway online software.

under grayanotoxin I stress, the growth and development of *S. litura* were significantly inhibited. Employing RNA-Seq, we have analyzed the transcriptome of *S. litura* to explore the molecular mechanisms responsible for the actions of grayanotoxin I. Many lipid metabolism-related genes responded to the treatment of grayanotoxin I, such as elevated expression of acyl-CoA desaturase, esterase E4, lipase H, and phospholipase A, and decreased expression of elongation of very long chain fatty acids protein, fatty acid-binding protein, acyl-CoA reductase wat, and pancreatic-like lipase. We also observed a significant reduction in the FFA level, activities of lipase, and HOAD after grayanotoxin I treatment. Based on these observations, we conclude that grayanotoxin I exerts its effects through, at least partly, modulating lipid metabolism-related gene expression in *S. litura*.

Lipids play crucial roles in the growth, development, and reproduction of insects. Fatty acid-derived wax esters, fatty alcohols, and hydrocarbons are essential components of the insect epidermis (*Teerawanichpan, Robertson & Qiu, 2010*). Very long-chain fatty acids serve as the precursors of sphingolipids and glycerolipids, two fundamental components

**Table 2  The KEGG pathway terms of lipid metabolism-related DEGs.**

| Pathway ID | Pathway | Level 1 | Level 2 | P-value | DGE ID | Up/down regulation |
|---|---|---|---|---|---|---|
| ko04212 | Longevity regulating pathway—worm | OS | Aging | 2.30E−06 | LOC111354773 | Up |
| ko04212 | Longevity regulating pathway—worm | OS | Aging | 2.30E−06 | LOC111355891 | Up |
| ko04212 | Longevity regulating pathway—worm | OS | Aging | 2.30E−06 | LOC111355893 | Up |
| ko04212 | Longevity regulating pathway—worm | OS | Aging | 2.30E−06 | LOC111352061 | Up |
| ko04212 | Longevity regulating pathway—worm | OS | Aging | 2.30E−06 | LOC111356581 | Down |
| ko04975 | Fat digestion and absorption | OS | Digestive system | 0.007337 | LOC111360381 | Down |
| ko03320 | PPAR signaling pathway | OS | Endocrine system | 0.030415 | LOC111355891 | Up |
| ko03320 | PPAR signaling pathway | OS | Endocrine system | 0.030415 | LOC111360381 | Down |
| ko03320 | PPAR signaling pathway | OS | Endocrine system | 0.030415 | LOC111355893 | Up |
| ko00073 | Cutin, suberin, and wax biosynthesis | M | Lipid metabolism | 1.99E−05 | LOC111354773 | Up |
| ko01040 | Biosynthesis of unsaturated fatty acids | M | Lipid metabolism | 0.011254 | LOC111355891 | Up |
| ko01040 | Biosynthesis of unsaturated fatty acids | M | Lipid metabolism | 0.011254 | LOC111350394 | Down |
| ko00062 | Fatty acid elongation | M | Lipid metabolism | 0.327476 | LOC111350394 | Down |
| ko01040 | Biosynthesis of unsaturated fatty acids | M | Lipid metabolism | 0.011254 | LOC111355893 | Up |
| ko00561 | Glycerolipid metabolism | M | Lipid metabolism | 0.559342 | LOC111348151 | Up |
| ko00073 | Cutin, suberin, and wax biosynthesis | M | Lipid metabolism | 1.99E−05 | LOC111352061 | Up |
| ko00073 | Cutin, suberin, and wax biosynthesis | M | Lipid metabolism | 1.99E−05 | LOC111356581 | Down |
| ko04152 | AMPK signaling pathway | EIP | Signal transduction | 0.268133 | LOC111355891 | Up |
| ko04146 | Peroxisome | CP | Transport and catabolism | 4.00E−05 | LOC111354773 | Up |
| ko04146 | Peroxisome | CP | Transport and catabolism | 4.00E−05 | LOC111352061 | Up |
| ko04146 | Peroxisome | CP | Transport and catabolism | 4.00E−05 | LOC111356581 | Down |

**Notes.**

OS, Organismal Systems; CP, Cellular Processes; M, Metabolism; EIP, Environmental Information Processing.

of cell membranes. Unsaturated fatty acids and fatty acid content are also crucial for the cold tolerance of insects (*Arrese & Soulages, 2010*). Furthermore, lipids serve as an essential energy source for insect activities (*Hannun & Obeid, 2002*; *Chertemps et al., 2007*). Due to the vital role lipids play in insects, lipid synthesis and lipolysis have become attractive targets for agriculture pest control. For instance, an *in vitro* enzyme kinetic experiment showed the pesticide spirotetramat bound to the carboxyltransferase (CT) domain of ACC and inhibited the fatty acid biosynthesis in *Myzus persicae*, *Spodoptera frugiperda*, and *Tetranychus urticae* (*Lümmen et al., 2014*). ACC is the rate-limiting enzyme in the initial step of fatty acid synthesis, responsible for insect lipid accumulation and epidermal function (*Ray, Wilkinson & Paul, 2018*). *Piper aduncum* (*Piperaceae*) essential oil, when delivered to insect thorax by micropipette, effectively depleted lipid content in fat body cells of brown stink bug *Euschistus heros* (*Heteroptera: Pentatomidae*), leading to the inhibition of bug development and reproduction (*Cossolin et al., 2019*). Similarly, *S. frugiperda* larvae, fed with corn leaf pieces immersed with citronella oil from *Cymbopogon winterianus*, increased glycogen, but decreased protein, lipid, and total sugar content leading to diminished reproduction (*Silva et al., 2016*). Our study observed a significant decrease in insect survival rate, suppression of larvae growth, and delay in pupation following grayanotoxin I treatment. Additionally, hemolymph FFA content and fat body

lipids were notably decreased. These phenotypes strongly suggested the involvement of lipid metabolism in the effects of grayanotoxin I on *S. litura.*

Lipase is an enzyme that catalyzes the hydrolysis of triglycerides into fatty acids and glycerol, playing a crucial role in the digestion and transportation of lipids. Insects possess several types of lipase, including pancreatic-like lipase, which hydrolyzes most dietary fats. Fatty acid-binding proteins (FABPs) are a group of small, soluble intracellular proteins responsible for efficient lipid trafficking and signaling within cells (*Furuhashi & Hotamisligil, 2008*). In our current study, we observed a significant decrease in FABP mRNA following grayanotoxin I treatment. FABPs are involved in regulating long-term memory, sleep, and lipid accumulation in insects (*Gerstner et al., 2011*). Two FABP subtypes, slFABP1 (MFB2) and slFABP2 (MFB1) were found in the midgut of *S. litura*, and they are known to participate in starving stress and body development (*Huang et al., 2012*). HOAD is a crucial enzyme involved in the beta-oxidation of lipids, which is responsible for the energy supply in insects. Grayanotoxin I treatment was found to suppress lipase and FABP activity, potentially disrupting the formation and trafficking of FFA in *S. litura.* Additionally, decreased HOAD activity may hinder fatty acid utilization and subsequent energy supply for the pest.

Our study uncovered a decrease in the elongation of very long chain (ELOVL) fatty acids elongase after grayanotoxin I treatment. ELOVL fatty acid elongase is primarily located on the endoplasmic reticulum (ER) and promotes the synthesis of C18-26 fatty acids from the C16 chain. ELOVL fatty acid elongases widely exist in different insects, such as *Bombyx mori*, *Locusta migratoria,* and *Ericerus pela Chavannes* (*Zuo et al., 2018*; *Zhao et al., 2020*; *Ding et al., 2022*). The very long chain fatty acids, including saturated and unsaturated fatty acids, are crucial sources of accumulated fat in the fat body of insects. Our present study found a significant decrease of ELOVL fatty acid elongase mRNA expression after grayanotoxin I treatment. Considering the important roles of ELOVL elongase in fat body development, we presumed that the effects of grayanotoxin I on *S. litura* growth and development might, at least partly, be related to the inhibition of ELOVL fatty acid elongase. Furthermore, our research revealed an increase in phospholipase A expression. Phospholipases hydrolyze phospholipids and participate in cell signaling pathways. The elevation of phospholipase A levels suggested the involvement of inflammation under grayanotoxin I stress.

In our studies, the gut poisoning of grayanotoxin I on *S. litura* was tested by diet mixed method according to the book "Standard Operation Practice for Pesticide Biological Activity Testing" by *Gu & Liu (2017)*. For the pesticide bioassay testing on *S. litura*, "diet mixed with insecticide" and "leaf dip feeding" were two commonly used methods for testing gut poisoning, while spray application was used for contact toxicity studies (*Gu & Liu, 2017*; *Bao et al., 2021*). In the lab bioassay of insecticide, the "diet mixed with insecticide" method was widely used because this method is simple, cost-effective, time-saving, and reliable. It is suitable for long-term medication and particularly appropriate for insecticides that are insoluble in water or have poor palatability (*Sarkar & Roy, 2017*; *Huang et al., 2021*; *Sun et al., 2022a*).

Besides *S. litura*, we have screened the insecticidal effects of grayanotoxin I on the Diamondback moth, beet armyworm, and budworm. *S. litura* was the most sensitive insect to grayanotoxin I, followed by Diamondback moth. Beet armyworm, and budworm were not sensitive to grayanotoxin I stress. Therefore, we selected *S.litura* as the target insect.

## CONCLUSIONS

The results of this study demonstrated that grayanotoxin I inhibited the growth and development of *S. litura*. The mechanisms might, at least partly, be related to the interference of lipid synthesis, lipolysis, and fat body development. These findings provide valuable insights into a new, environmentally-friendly plant-derived insecticide, grayanotoxin I, to control the spread of *S. litura*.

### Funding

This work was supported by the Hunan Key Laboratory of the Research and Development of Novel Pharmaceutical Preparations; the Hunan Provincial Key Laboratory of the TCM Agricultural Biogenomics; the ''14th Five-Year Plan'' Application Characteristic Discipline of Hunan Province (Pharmaceutical Science); the Provincial Key R & D projects of Hunan Provincial Science and Technology Department under Grant No. 2022SK2074 and the ESI Discipline Special Project of Changsha Medical University under Grant No. 2022CYY001 and 2022CYY002. The funders had no role in study design, data collection and analysis, decision to publish, or preparation of the manuscript.

### Grant Disclosures

The following grant information was disclosed by the authors:
Hunan Key Laboratory of the Research and Development of Novel Pharmaceutical Preparations; the Hunan Provincial Key Laboratory of the TCM Agricultural Biogenomics.
''14th Five-Year Plan'' Application Characteristic Discipline of Hunan Province (Pharmaceutical Science).
The Provincial Key R & D projects of Hunan Provincial Science and Technology Department: 2022SK2074.
ESI Discipline Special Project of Changsha Medical University: 2022CYY001, 2022CYY002.

### Competing Interests

The authors declare there are no competing interests.

### Author Contributions

- Yi Zhou performed the experiments, prepared figures and/or tables, and approved the final draft.
- Yongmei Wu performed the experiments, prepared figures and/or tables, and approved the final draft.

- Rong Fan performed the experiments, prepared figures and/or tables, and approved the final draft.
- Jiang Ouyang performed the experiments, prepared figures and/or tables, and approved the final draft.
- Xiaolong Zhou performed the experiments, prepared figures and/or tables, and approved the final draft.
- Zibo Li analyzed the data, prepared figures and/or tables, and approved the final draft.
- Muhammad Usman Janjua analyzed the data, authored or reviewed drafts of the article, and approved the final draft.
- Haigang Li analyzed the data, authored or reviewed drafts of the article, and approved the final draft.
- Meihua Bao conceived and designed the experiments, authored or reviewed drafts of the article, and approved the final draft.
- Bin-sheng He conceived and designed the experiments, authored or reviewed drafts of the article, and approved the final draft.

## Microarray Data Deposition

The following information was supplied regarding the deposition of microarray data:

The sequences are available at the Sequence Read Archive (SRA): PRJNA957576.

## Data Availability

The raw measurements are available in the Supplementary Files.

## Supplemental Information

Supplemental information for this article can be found online at http://dx.doi.org/10.7717/peerj.16238#supplemental-information.

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
