# Peer review of "Transcriptome analysis unveils the mechanisms of lipid metabolism response to grayanotoxin I stress in Spodoptera litura"

_PeerJ, doi:10.7717/peerj.16238_

## Round 0.1 · original submission · Major Revisions

The authors need to include all suggestions given by reviewers before re-submission.

Reviewer 1 ·

Basic reporting

The following suggestions can be used to improve the language used in the manuscript:

Line 47: Pieris japonica should be italicized
Line 120: Is this supposed to be OCT compound? (and not OTC compound)
Line 183: change “statistic” to “statistical”
Line 424: “hotmap” should be “heatmap”

Part of Figure 3 also seems to be cut off.

Experimental design

Section 2.5 does not state whether the transcriptome was aligned to the genome reference to obtain read counts or if a de novo assembly was used. The software used to conduct the analysis should be mentioned.

Validity of the findings

This reviewer questions the validity of the findings based on the following observations. If the authors can provide a reasonable reason for this discrepancy, then this comment can be disregarded.

This study is similar to a study described in a preprint that was also written by the authors (https://papers.ssrn.com/sol3/papers.cfm?abstract_id=4090282). The main difference in the methodology between both papers is that this one is testing the effects of a commercially available grayanotoxin I (line 89) while the preprint is testing the effects of a P. japonica ethyl acetate extract they prepared, which has a mixture of grayanotoxin and one or more other compounds (e.g. asebotoxin I). Even though different extracts were tested in both manuscripts, the results in some cases are the same:

285 DEGs (151 Up-regulated, 134 down-regulated)

18.2% survival rate

9.3% weight reduction

“concentrations reaching 66%, 83%, and 72%”

… etc.

Based on random chance alone, it is highly improbable that the results of both experiments would be identical, and it brings into question the validity of the findings in this manuscript.

·

Basic reporting

1. The introduction is excessively brief. Please provide more comprehensive details about grayanotoxins. Additionally, in Line 79, when referring to the sentence "Grayanotoxins I, the main active ingredient in Pieris japonica..." What is meant by "ingredient"? Is it in the plant extract? What type of extract? Please provide further specifics.
2. Some acronyms are defined in the abstract but are not described in the main text. For example, in Line 84, please define HOAD and ACC.
3. Furthermore, as a non-native speaker, the text appears professional to me, with only minor points that need attention.
- Change "Pieris Japonica" to "Pieris japonica" (scientific notation) throughout the text.
- In Line 69, what is meant by "explosive"?
- Line 98: Do not abbreviate scientific names in the headings.
- Line 142: Reference [10] is not in the correct format.
- Line 195: Please change "worm" to "larvae" here and throughout the text.
- Line 285: Please define CT.
- Lines 286-287 and 290-292: Put scientific names in italics.
- Line 299: Insects do not have a pancreas, please refer to this enzyme as pancreatic-like lipase.
- Line 411: Change "&" to "and".
- Line 424: Change "hotmap" to "heatmap".

Experimental design

1. Lines 145–155: I have concerns about the qPCR experiments:
- Only one reference gene was used. According to the MiQE guidelines (DOI: 10.1373/clinchem.2008.112797), at least two reference genes must be used to properly quantify gene expression by qPCR.
- The authors did not mention if they performed a melting curve analysis in the experiments to confirm the amplification of a single fragment.
- No negative controls, such as non-transcribed RNA, were mentioned.
- Please put the primer sequences in a table and provide more details, such as the target gene regions they amplify.
2. Line 105: Please define what matrine is and explain why it was chosen as the positive control.
3. Lines 131-132: Provide a better description of the sequencing process. Was it paired-end or single-end? etc.
4. Line 141: Why did the authors choose these four genes to validate the RNAseq by qPCR?
5. Line 169: Is ELISA the best method to quantify the levels of FFA? Please provide justification.
6. Line 111: Change "lower concentrations" to "sublethal concentrations".
7. Line 410: Provide a more detailed description in the figure legends, avoid abbreviations, and include information on statistics. Figure legends should be self-explanatory.

Validity of the findings

1. The authors solely focus the results and discussion of the paper on lipid metabolism, but the transcriptome should provide other insights that would help explain the effects of GI in S. litura, such as hormone biosynthesis or chitin metabolism.
2. Where is the statistical analysis for the data in Figures 1A and 1C?
3. Line 233: The difference in ACC levels was not significant; please rephrase the sentence.
4. Line 270: In fact, the RNAseq reveals the modulation of gene expression in response to treatment. Gene modulation is not solely a direct response to GI action but also a secondary stress response caused by homeostatic imbalance.
5. Lines 311-316: I don't see how the data can support this statement. Please provide a detailed explanation or modify it accordingly.
6. Line 323: Please change "...suppression effects by disrupting lipid synthesis..." to "...suppression effects, probably by disrupting lipid synthesis...".

Additional comments

Could these results be extrapolated to other species, such as the significant invasive species Spodoptera frugiperda? The authors can enhance the discussion by incorporating this information.

·

Basic reporting

This study investigated the effects of plant toxin on the tobacco cutworm pest and its underlying mechanisms. Grayanotoxin I significantly suppressed survival, growth, development, and pupation. RNA-Seq analysis identified 285 differentially expressed genes (DEGs), including genes related to lipid metabolism. Enrichment analysis revealed the involvement of these DEGs in cuticle development, worm longevity, and fat digestion and absorption. The toxins also inhibited the levels of certain enzymes and molecules involved in lipid metabolism. This study showed that grayanotoxin I interferes with lipid synthesis, lipolysis, and fat body development, providing insights for environmentally-friendly pest control methods.

General comments:
Please check for the capital letters (initial) used for certain chemical compound. Please be consistent. Additionally, please write the scientific names of organisms in Italic. Please thoroughly check it. Furthermore, please mention the full forms before using the abbreviation. Overall, I would encourage the authors to improve the English language.

What would be the efficacy of the toxins when applied as a spray? It should be better if that is at least discussed.

Please cite more papers in the realm of development of “insecticide” from the plant source and their effects on the insects’ life history.

Experimental design

Line 105: matrine comes all of a sudden. Please introduce it.

Line 108: Was the calculation done in a dry weight basis or wet weight basis? Please mention it.

Line 111: Why lower concentrations were chosen here? Please mention the rationale behind it. Lower concentrations could also lead to hormesis, please see here https://www.nature.com/articles/s41514-017-0013-z

Line 120: What is a OTC compound? Please mention the full form before using the abbreviation.

Line 162: Why was hemolymph allowed to coagulate?

Validity of the findings

Line 285-293: In the references cited, how was the “chemicals” applied to the insects? Was the mode of application systemic or topical? Please mention it.

Line 299: Insects do not have a pancreas. Please rewrite the sentence for better clarity.

Additional comments

Line 69: Please also include the common name of the pest. The word “explosive” sounds very strange in this context.

Line 72-73: There has to be clearly mentioned the difference between synthetic chemical pesticide and the “pesticide” from the plant extracts.

Line 77: Please add a reference to it.

Line 78: Please rewrite the sentence. “mechanisms of Pieris Japonica on agricultural pests are still largely”. Replace the plant with the toxin

Line 79: Grayanotoxins I is incorrect. Please remove the s

Line 84: Please mention the full form of HOAD, ACC

Reviewer 4 ·

Basic reporting

Regarding Figure 1d and Figure 4b-f, it is important for the authors to clarify the meaning of the * symbol used in these figures. This can be done by including a corresponding legend or caption that explains the significance levels associated with the * symbol. Providing clear and concise explanations will help readers interpret the results accurately.

Experimental design

The authors in this study used a p-value threshold of less than 0.05 as a criterion for determining statistical significance. The use of adjusted p-values to account for multiple comparisons is a common practice in differential expression analysis. Can you please explain why you don't use the adjusted p-values for the multiple comparisons present in the DEG analysis?

Validity of the findings

As long as the authors can demonstrate the use of p-values rather than adjusted p-values, the results of this study are promising.

---

## Round 0.2 · Minor Revisions

The authors need to include responses to all reviewers before re-submission.

Reviewer 1 ·

Basic reporting

Line 71: italicize species name
Line 89: needs restructuring
Line 166: “sequence” should be plural and should “mapping” be ‘mapped’?
Line 303: needs restructuring
Line 323: italicize "in vitro"
Line 369: withi?

The overall English has improved throughout.

Experimental design

The recommended revisions were made.

Validity of the findings

In response to my previous comment on the findings being published in a now-retracted preprint with a different extract used, the authors claimed that there was a data mix-up. Mix-ups are understandable when working with large datasets, however, I will leave it to the editor's discretion whether more evidence is needed to substantiate this claim.

For example, an RNA-seq experiment was conducted in the pre-print. If there was a mix-up with the data from this study, then there should be a corresponding NCBI BioProject raw data accession for the RNA-seq data used in the pre-print (or at least the raw data from that study should be stored in an online repository). Or, if there was natural products extraction to identify the grayanotoxin I from the Pieris japonica extracts, this could be provided in the methods section or as supplementary data (NMR data, chromatographic separation data etc).

·

Basic reporting

All my suggestions were promptly addressed. No further comments

Experimental design

No comment

Validity of the findings

No comment

Additional comments

No comment

·

Basic reporting

The manuscript has undergone revisions; however, I still recommend that the authors seek assistance to improve their English grammar. While I have made some corrections, comprehensive proofreading is necessary.

Experimental design

no comment

Validity of the findings

no comment

Additional comments

• Line 42: Including the insect's common name in parentheses would be advisable.
• Line 46: Replacing "stomach" with "gut" might improve the phrasing.
• Line 50: Consider replacing "tested" with "recorded" for better accuracy.
• Line 59: "Worm longevity" sounds awkward. Perhaps consider using "caterpillar" or "larvae longevity" instead.
• Line 71: Ensure that the species name is in italics; please make this correction.
• Line 80: Omit "etc." and conclude the sentence with "and" for clarity.
• Line 123: Kindly provide the source of matrine and its chemical compound class.
• Maintain consistency in the spacing between Celsius/percentage and the corresponding numbers throughout the manuscript.
• Line 226: Please, avoid referring to matrine as a drug. Maintain consistent terminology.
• Line 374: Organisms' common names should not be italicized. Please rectify this.
• Figure 1 C: Change "worm" to "larvae" or "caterpillar" for consistency throughout.

Reviewer 4 ·

Basic reporting

The basic reporting is clear. The authors used professional English in the paper. The literature references and field background are provided. The paper has a good structure of paragraph, tables and figures.

Experimental design

Since the manuscript was refined based on the reviewers' comments. The experimental design has been improved. Specifically, the research question is well defined and meaningful. The methods in this paper were described with sufficient details.

Validity of the findings

The findings in this refined paper are relatively promising. Since the authors used appropriate methods in this study, the results and conclusions are reasonable. The conclusions are well stated and linked to original research question.

Additional comments

I recommend to accept this paper.

---

## Round 0.3 · accepted · Accept

This revised article is now good to accept for publication.

Reviewer 1 ·

Basic reporting

The recommended revisions were made.

Experimental design

no comment

Validity of the findings

Underlying data from the previous study they conducted was provided, so that the findings attributed to this study seem valid.

Additional comments

no comment